# Exploring awareness of hearing loss and ear health in Jordanian adults

**Yazan Gammoh**[1]*, **Rama Alasir**[2], **Laila Qanawati**[2]

**1** Faculty of Allied Medical Sciences, Department of Optometry Science, Al-Ahliyya Amman University, Amman, Jordan, **2** Faculty of Allied Medical Sciences, Department of Audiology and Speech Pathology, Al-Ahliyya Amman University, Amman, Jordan

* gammohyazan@yahoo.com

## Abstract

### Objective

To assess the awareness about hearing loss and ear health among adults in Jordan.

### Methods

A cross-sectional study was conducted where a questionnaire was filled from the month of November to the month of December of the year 2023, to assess the level of awareness about hearing loss and ear health. The participants included were Jordanian adults (age $\geq$ 18 years) residing in the North, Middle and South of Jordan.

### Results

Data from 333 participants (54.1% men) were analyzed. Participants between 18 and 28 years of age comprised 29.7% of the sample population. More than half of the participants (52.6%) held a university degree. Overall percentage of correct responses was 83%. Women, postgraduate degree holders, and participants diagnosed with hearing loss had an average of 11.96±1.47, 12.65±1.59 and 11.70±1.69 correct answers, respectively. The highest correct response received (97.6%) was for: hearing aids need to fit accurately to provide the maximum benefit. Furthermore, 97% of the sample correctly acknowledged that sudden hearing loss is an emergency and requires an immediate audiological assessment. The main misconception was that a deaf–mute cannot speak because of defects in the vocal tract, with only 39.3% of the sample providing a correct response. The other two misconceptions were: cotton buds are necessary for ear cleaning and are the safest means, and that ear drops are sufficient to treat earache, with 78.1% correct responses for each statement. Participants with higher level of education had higher odds of answering the questions correctly, with limited role observed for gender, prior diagnosis of hearing loss and a family history of hearing loss.

### Conclusions and relevance

Majority of the adults surveyed provided a correct answer to the hearing loss and ear health survey. While most of the sample population were aware that a sudden loss of hearing is

**Data Availability Statement:** All relevant data are within the manuscript and its Supporting Information files.

**Funding:** The author(s) received no specific funding for this work.

**Competing interests:** The authors have declared that no competing interests exist.

considered an emergency, only one third knew that defects in vocal cords do not play a role in deafness/muteness. The study highlights the need of public education on causes of hearing loss and measures needed to prevent the onset of hearing loss, with emphasis on methods for caring of ear health.

## Introduction

Hearing loss (HL) is considered one of the most prevalent sensory impairment with approximately 6.8% of the world population suffering from a disabling hearing impairment [1]. In the Eastern Mediterranean region, more than 22 million people are estimated to have HL, and this number is expected to more than double by the year 2050 [2]. The impact of HL is compounded by inequality in geographic distribution of HL, where approximately 80% of people with HL residing in low- and middle-income countries, such as Jordan [3]. It is estimated that in Jordan, 1.37% of the newborns have permanent conductive HL (which refers to hearing loss to the outer and middle ear) and sensorineural HL (which refers to hearing loss to the inner ear, the auditory nerve, or the brain); approximately three times the prevalence of newborns in some neighboring countries [4]. Among industrial workers in Jordan who are exposed to noise, noise-induced HL is found among one quarter of the workers [5]. While a recent study that assessed HL among 1000 Jordanians over 12 years of age has shown that the prevalence of HL in Jordan is approximately 6.3% [6].

Many factors and condition may cause HL, such as otitis media [7], exposure to loud noise [8], and infectious diseases such as mumps, measles, rubella, and meningitis [9]. Providing ear health and care education to the public has been shown to limit, delay or avoid the consequent HL due to the above-mentioned factors [10]. In addition, prevention and early management of conditions that may cause HL can be influenced by the general population's knowledge about ear health and HL [11,12].

Despite the need for ear health education, there is a paucity of information about the general population knowledge about HL and ear health [10–12]. This is of importance as HL may impact the economic, social, and psychological state of the person with HL [10,13]. In addition, as persons with HL may feel the stigma of HL, it can lead to denial of use of assistive listening devices limiting the person's ability to communicate and interact with their environment [13]. A survey conducted on adolescents and young adults in the United States of America has shown that only 8% of participants viewed hearing loss as a significant concern. In contrast, other health issues, such as depression and sexually transmitted diseases, were regarded as very serious problems by 44% and 50% of respondents, respectively [14]. The researchers have also noted that participants did not consider HL as a serious problem, and did not realize its effect on quality of life. It is of interest to note that HL has a profound negative impact on quality of life as it may lead to depression and limit social communication [15]. In a study conducted on young adults in Jordan, HL was considered less detrimental compared to depression [16]. A study conducted on general practitioners and pediatricians revealed that these physicians have low of awareness regarding HL in children, despite the extensive medical education and training they received [17]. Despite the above information about awareness of HL, there is limited evidence regarding the general population awareness of HL in Jordan.

It has been shown that early intervention is crucial in reducing the impact of HL and its consequences, which is important given the lower cost of early intervention to management once the conditions that may lead to HL occur [18]. Many cases of HL could be prevented

through public health awareness and interventions [19]. It has been shown that raising the level of awareness of the population regarding HL, combined with concrete public health initiatives and collaboration with policymakers and stakeholders will allow for early intervention leading to reduction in HL [20,21]. However, the stigma of hearing loss and the misconceptions of the people at the community level hinders the efforts to promote preventive strategies for HL [22].

With limited availability of information about awareness of ear health and HL from the general adult population; especially from middle-income countries such as Jordan [23], the study aimed to investigate the awareness of adults in Jordan regarding HL and ear health.

## Materials and methods

The Scientific Research Ethics Committee at Al-Ahliyya Amman University has granted the researchers ethical approval to conduct the current study (IRB: AAU/2/4/2023-2024). The study adhered to the Declaration of Helsinki and the participants provided written informed consent. No monetary compensation was offered to the participants and they had the right to withdraw from the study at any time.

The study utilized a cross-sectional design where a validated questionnaire was filled by Jordanians who are 18 years of age and over. The questionnaire collection period started from November, 20th, 2023 and ended on December, 20th, 2023. The study sample population comprised of participants from universities and workplaces across the Northern, Middle, and Southern regions of Jordan [24]. A one-dimensional systematic sampling approach was utilized where every third person entering a building in an establishment was approached, thus ensuring avoidance of sampling bias, ensuring representativeness as family members or relatives may enter such establishments as a group. Avoiding sampling from family members groups would lower the risk of the role of consanguinity as observed among Jordanians [25]. Inclusion criteria included: adults aged 18 years and above, ability to understand the questionnaire and provide responses to the questions and consenting to participate in the study. Persons who were under the age of 18 years and those who did not consent to participate were excluded. Only one researcher had administered the questionnaire to avoid inter-user sampling error. The researcher was trained by two audiology and speech pathology professionals to ensure the proper administration of the questionnaire. The professionals observed the administration of the questionnaire on a group of twenty adults who were not included in the study and the data collection was started once the professionals approved the readiness of the researcher to administer the questionnaire. After the data collection was completed, all questionnaire were inspected to ensure all data were filled. Any questionnaire with missing data were excluded from inclusion in the data analysis, leading to a total of 333 responses included in the analysis and 12 excluded (exclusion rate of 3.5%).

### Awareness about hearing loss questionnaire

The questionnaire adopted in the study was previously utilized in literature to assess the awareness of the general population about hearing loss and ear health [11,12]. This questionnaire contained the following questions/statements which required either a "true" or a "false" response:

1. It is possible to diagnose deafness in infants shortly after birth? (correct response: True)

2. A deaf–mute cannot speak because of defects in the vocal tract. (correct response: False)

3. Hearing loss may cause attention deficits thus reducing school performance. (correct response: True)

4. Cotton buds are necessary for ear cleaning and are the safest means. (correct response: False)

5. Ear drops are sufficient to treat earache. (correct response: False)

6. Otomycosis (itchy ears) can be contracted at the swimming pool. (correct response: True)

7. Drug abuse does not provoke auditory hallucinations or modifications of hearing quality. (correct response: False)

8. Hearing aids need to fit accurately to provide the maximum benefit. (correct response: True)

9. Kisses or slaps on the ears do not cause hearing problems. (correct response: False)

10. Listening to music for more than 3 hours a day using earphones may cause permanent hearing loss. (correct response: True)

11. There are no tables recommending a reduction in the duration of exposure to high-intensity noises. (correct response: False)

12. Irritating perception of sound (e.g. hearing metallic voices) and/or a reduction in hearing clarity (such as a sensation of having cotton wool in the ears) require medical advice. (correct response: True)

13. Sudden hearing loss is an emergency and requires an immediate audiological assessment. (correct response: True)

14. Age-related hearing loss may affect behavior. (correct response: True)

The questionnaire was originally adapted from World Health Organization materials by Di Berardino and colleagues (2013) [11,12] and applied to an Italian population. The same questionnaire has been translated to Arabic and was used on a Saudi population sample by AlShehri and colleagues (2019) [11,12]. The current study adopted the same forward and backward translation method used in the Saudi study [11,12]. The questionnaire was translated from English to Arabic (forward translation) by two independent speech and language specialists. Then backward translation from Arabic to English was conducted by another set of two independent speech and language specialists. The authors compared the original version and the translated version to ensure consistency and manage any differences.

The recommendations set by the Strengthening of Reporting in Observational studies in Epidemiology (STROBE) statement were followed in the methods adopted by the study and the results obtained. Assuming a confidence interval level at 95%, with a ± 5% margin of error and a replacement rate of 5%, a sample size of 405 participants was considered representative of the Jordanian adult's population [26]. Data were entered using Microsoft Excel Spreadsheets and data analysis was performed using SPSS software version 25 (IBM Corporation, Armonk, NY, USA). Numbers and percentages were used for descriptive statistics. The number of correct answers to the questionnaires was counted and the average of correct answer for each independent variable (for example, men and women) were calculated. Independent samples t-test were used for assessing the difference in the average number of correct results categories with two independent variables such as men and women, or persons diagnosed with hearing loss versus persons without hearing loss. While one-way analysis of variance, with Bonferroni correctio, were used to assess the difference in the average number of correct results among more than two independent variables such as age groups. Binary logistic regression analysis was utilized to calculate the Odds ratio and 95% confidence interval for correct answers to

each statement in the questionnaire. A P value less than 0.05 was considered to be statistically significant.

## Results

A total of 450 Jordanian adults (age 18 years and above) were invited to participate in the study. However, only 333 responses were included as the participants consented and had full questionnaire with no missing data (a response rate of 74%). Men comprised more than half of the participants (54.1%), and 29.7% of the participants in the study population were between 18 and 28 years of age. To categorize the population sample, the participants were asked whether they were diagnosed with a hearing loss by a healthcare professional and have been divided into two categories: participants who reported a diagnosis of hearing loss or a who were not diagnosed with a hearing loss. Few of the participants reported being diagnosed with hearing loss (7.5%, n = 25), or a having a positive family history of hearing loss (15.4%, n = 51). Table 1 details the demographic profile of the participants.

Table 2 details the correct response to each statement in the questionnaire and the percentage of correct responses per statement. Overall percentage of correct responses was 83% (n = 2760 with the least number of correct responses (39.3%, n = 131) were for the statement "A deaf–mute cannot speak because of defects in the vocal tract", and the highest (97.6%, 325) were for "Hearing aids need to fit accurately to provide the maximum benefit".

In terms of the percentage of correct responses to the four domains of the questionnaire, the "Diagnostic Delay" domain received the most correct number of responses (92.4%, n = 308). On the other hand, the "Infant Hearing Loss" domain was correctly answered by 72.7% (n = 242) of the participants. The other two domains "Cleaning and Treating" and "Physical Agents and Overexposure" received 85.6% (n = 285) and 79.5% (n = 265) correct responses, respectively.

The average number of correct answers by the participants was 11.61 answers (± 1.72). All demographics variables showed a statistically significant differences in the mean of correct answers per category in each variable. Table 3 demonstrates the average number of correct answers per each demographic variable. Women had more correct answers than men (p = 0.001). While participants between the age of 18 to 28 years hand answered the

**Table 1. Demographic profile of the study population (n = 333).**

| Variable | Category | Number | Percentage |
|---|---|---|---|
| **Gender** | Men | 180 | 54.1% |
| | Women | 153 | 45.9% |
| **Age group (years)** | 18–28 | 99 | 29.7% |
| | 29–39 | 92 | 27.6% |
| | 40–50 | 70 | 21% |
| | 51–61 | 45 | 13.5% |
| | ≥62 | 27 | 8.1% |
| **Level of education** | Secondary school | 42 | 12.6% |
| | High school diploma | 82 | 24.6% |
| | University degree | 175 | 52.6% |
| | Postgraduate degree | 34 | 10.2% |
| **Diagnosis of hearing loss** | Yes | 25 | 7.5% |
| | No | 308 | 92.5% |
| **Family history of hearing loss** | Yes | 51 | 15.4% |
| | No | 282 | 84.6% |

**Table 2. Questionnaire components, correct response per statement and participants' responses (n = 333).**

| Domain | Statement | Correct response | Percentage of correct responses |
|---|---|---|---|
| Infant hearing loss | 1. It is possible to diagnose deafness in infants shortly after birth | True | 84.1% (n = 280) |
| | 2. A deaf–mute cannot speak because of defects in the vocal tract | False | 39.3% (n = 131) |
| | 3. Hearing loss may cause attention deficits thus reducing school performance | True | 94.6% (n = 315) |
| Cleaning and treating | 4. Cotton buds are necessary for ear cleaning and are the safest means | False | 78.1% (n = 260) |
| | 5. Ear drops are sufficient to treat earache | False | 78.1% (n = 260) |
| | 6. Otomycosis (itchy ears) can be contracted at the swimming pool | True | 91.6% (n = 305) |
| | 7. Drug abuse does not provoke auditory hallucinations or modifications of hearing quality | False | 82.6% (n = 275) |
| | 8. Hearing aids need to fit accurately to provide the maximum benefit | True | 97.6% (n = 325) |
| Physical agents and overexposure | 9. Kisses or slaps on the ears do not cause hearing problems | False | 85.6% (n = 285) |
| | 10. Listening to music for more than 3 hours a day using earphones may cause permanent hearing loss | True | 72.4% (n = 241) |
| | 11. There are no tables recommending a reduction in the duration of exposure to high-intensity noises | False | 80.4% (n = 268) |
| Diagnostic delay | 12. Irritating perception of sound (e.g. hearing metallic voices) and/or a reduction in hearing clarity (such as a sensation of having cotton wool in the ears) require medical advice | True | 93.1% (n = 310) |
| | 13. Sudden hearing loss is an emergency and requires an immediate audiological assessment | True | 97% (n = 323) |
| | 14. Age-related hearing loss may affect behavior | True | 87.1% (n = 290) |
| **Overall percentage of correct responses** | | | **83% (n = 276)** |

questionnaire more correctly compared to other age groups (p = 0.034). Participants with higher education level had more correct answers compared to those with lower education level (p<0.001). Participants who were diagnosed with hearing loss or had a family history of hearing loss answered more questions correctly compared to their peers (p = 0.001, p = 0.007; respectively).

**Table 3. Mean ± standard deviation of correct answers among the study sample population (n = 333).**

| Variable | Category | Mean ± Standard Deviation of correct answers | P* |
|---|---|---|---|
| **Gender** | Men | 11.32 ± 1.87 | 0.001 |
| | Women | 11.96 ± 1.47 | |
| **Age group (years)** | 18–28 | 11.96 ± 1.79 | 0.034 |
| | 29–39 | 11.72 ± 1.68 | |
| | 40–50 | 11.47 ± 1.85 | |
| | 51–61 | 11.18 ± 1.48 | |
| | ≥62 | 11.07 ± 1.41 | |
| **Level of education** | Secondary school | 10.67 ± 1.51 | <0.001 |
| | High school diploma | 11.44 ± 1.67 | |
| | University degree | 11.72 ± 1.70 | |
| | Postgraduate degree | 12.65 ± 1.59 | |
| **Diagnosis of hearing loss as reported by the participants** | Yes | 11.70 ± 1.69 | 0.001 |
| | No | 10.56 ± 1.76 | |
| **Family history of hearing loss** | Yes | 11.72 ± 1.71 | 0.007 |
| | No | 11.02 ± 1.67 | |

*Significance was set at p<0.05. One-way analysis of variance was used for the age group and level of education variables, independent sample t-test was utilized for other variables.

**Table 4. Assessment of the role of demographics variables in providing a correct answer for the statements in the questionnaire using binary logistic regression analysis (statistically significant results are only shown, n = 333).**

| Statement | Variable | Odds Ratio | 95% Confidence Interval | P |
|---|---|---|---|---|
| 2. A deaf–mute cannot speak because of defects in the vocal tract | Age<br>Level of education<br>Family history of hearing loss | 0.76<br>1.78<br>3.43 | 0.62–0.94<br>1.29–2.45<br>1.48–7.94 | 0.009<br><0.001<br>0.004 |
| 3. Hearing loss may cause attention deficits thus reducing school performance | Diagnosis of hearing loss | 5.89 | 1.37–25.39 | 0.017 |
| 4. Cotton buds are necessary for ear cleaning and are the safest means | Level of education<br>Diagnosis of hearing loss | 1.79<br>6.27 | 1.25–2.57<br>2.36–16.65 | .002<br><0.001 |
| 5. Ear drops are sufficient to treat earache | Level of education<br>Gender | 1.91<br>2.96 | 1.35–2.69<br>1.56–5.63 | <0.001<br>0.001 |
| 6. Otomycosis (itchy ears) can be contracted at the swimming pool | Gender | 4.33 | 1.66–11.31 | 0.003 |
| 7. Drug abuse does not provoke auditory hallucinations or modifications of hearing quality | Level of education | 1.87 | 1.27–2.74 | 0.001 |
| 9. Kisses or slaps on the ears do not cause hearing problems | Level of education | 1.99 | 1.33–3.02 | 0.001 |

Table 4 details the statistically significant relationship (Odds Ratio ± 95% confidence interval) between the demographics' variables and the correct response to the questions included in the questionnaire using binary logistic regression. Only questions 2, 3, 4, 5, 6, 7 and 9 showed that certain demographic variables may influence the odds of answering the questions correctly. Participants with higher level of education had higher odds of answering questions 2, 4, 5, 7 and 9. Female had higher odds of answering questions 5 and 6 correctly. Participants with hearing loss had higher odds of answering questions 3 and 4 correctly. Participants with family history of hearing loss had higher odds of answering question 2 correctly.

## Discussion

A recent study conducted on the awareness of Jordanian public about audiology and speech language pathology professions have revealed a low level of awareness about these professions [23]. The authors also called for the need to raise the public knowledge of audiological services, which should include knowledge about ear care and ear health as it was not addressed in their study [23]. There is limited literature regarding the public knowledge of HL and ear health in the Middle East with no data available from public in Jordan. Thus, this study aimed to assess the level of knowledge among the Jordanian adult population about matters related to ear health and hearing loss.

This study utilized a 14-item questionnaire that has been developed and employed firstly on the public in Italy [12], and later translated to Arabic and administered to the public in Saudi Arabia [11]. Most of the participants (83%) provided correct responses to the questions included in the survey, which is in slightly higher compared to data obtained from Italy (80%) and Saudi Arabia (76%) [11,12]. The higher correct response rate observed in the current study could be attributed to the high level of education attained by the participants, as well as it included only adults who may be more aware of HL and ear health compared to children under the age of 18 years. The study conducted in Saudi Arabia has shown that children are less likely to provide correct responses compared to adult, where the difference in the correct response rate was statistically significant [11]. However, none of the studies conducted in Italy and Saudi Arabia investigated the effect of level of knowledge on the number of correct responses provided by the public. It has been shown that higher level of education also affects knowledge about other diseases such as diabetes and diabetic retinopathy, thus our results are in alignment with other studies conducted in Jordan [27]. A recent study that investigated the

level of knowledge and awareness of hereditary HL indicated that a higher level of knowledge and education on HL, especially the hereditary type is detrimental in reducing its impact and occurrence [28]. This solidifies the role of raising the level of education of the general population in Jordan on causes and consequences of hearing loss which would lead to reduced occurrence [29].

In terms of gender, women have demonstrated a higher level of knowledge about HL and ear health compared to men, especially in the role of swimming pools in otomycosis and the role of ear drops in the treatment of earache. The relation between gender and level of knowledge has also been demonstrated among the studies in Italy and Saudi where women have provided more correct answers compared to men [11,12]. This was attributed to the role of women in raising children and the family in the household, which is still observed in the Jordanian culture [30]. Family obligations and role expectations of females/mothers in the Jordanian context endorses women as primary caregivers, furthermore, chronic illness impacts the health and quality of life of caregivers, with mothers typically taking on the role of primary caregivers, especially in the Middle East [31]. This calls for the need of public health campaigns tailored for men to raise their level of awareness about ear health and care which would provide a positive impact on reducing the occurrence of HL in the population.

Having a family history of HL also affects the individual's response to the questionnaire. This is of particular importance to the question "A deaf–mute cannot speak because of defects in the vocal tract" which received the lower rate of correct responses (39.3%). It has also been observed that a low response rate to this question was observed in Saudi Arabia, but not in the study conducted in Italy [11,12]. The study in Saudi Arabia did not provide a justification for this misconception, however we assume this is due to the fact that it is less common for public to be aware of the anatomy and physiology of the human body, specifically speech and hearing anatomy and physiology. Unless individuals who filled the questionnaire were doctors or health professionals it not easy to have this kind of knowledge as its not common knowledge or even discussed in social gatherings. There is an apparent low level of knowledge among the Jordanian general population about other related medical issues, such as medications, where many do not understand the impact of self-medication on the human body which translates to poor knowledge of the human physiology [26]. This prompts the need for public health education where campaigns led by individuals who have a family history of HL can promote awareness to avoid such misconceptions among the society.

Among the study's participants 7.5% mentioned that they have been diagnosed with HL. The current study did not assess HL among the participants, rather depended on self-reporting which would raise the question about the robustness of the results obtained. Ensuring representation of individuals with HL or family history of hearing impairments can be challenging due to their lower numbers compared to those with normal hearing and no family history of HL. This is especially more challenging in Jordan where the prevalence of HL is low with only 63 persons out of 1000 having HL [6]. Being diagnosed with HL has been linked to a higher probability of answering correctly to the question related to the use of cotton buds, another misconception in the current study that needs to be addressed. Cotton buds are recommended mainly for cleaning the external part of the ear or absorbing excess water if it enters the ear [32]. However, cotton buds have been used by many individuals to clean the ear canal from the excess wax which has been shown to cause clinical complications [10,32]. It has been shown that only half of the population in South Africa are aware of the role of cotton buds in ear infections, however there is a lack of information from Jordan [33]. The study in South Africa has called for the need of intervention strategies that would also allow for achieving the United Nations' Sustainable Development Goals (SDGs), to address the detriments of health including social aspects [34]. There is robust evidence that healthcare providers play an

important role in raising patient awareness about health issues [35]. It is recommended that family physicians, nurses and community health workers provide advice to families and individuals about the proper use of ear buds and avoiding misuse. Although primary care physicians have been shown to demonstrate low level of knowledge about HL in children, they demonstrated positive attitude towards HL in children which would contribute to raising the public knowledge on proper use of cotton buds [36].

The question related to accurate fitting of hearing aids received the higher number of correct responses (97.6%), which interestingly was the question answered correctly by most of the public in Saudi Arabia (93%) [11]. Though in Italy, where the sample population of the study were older, have demonstrated a low level of knowledge about the need to fitting hearing aids accurately (35.85%) [12]. It has been speculated that this discrepancy in knowledge observed among the Italian population could be attributed to the elderly, who constituted a large proportion of the study sample, not being as aware as the younger adults of the importance of hearing aids as they were developed towards the later part of the twentieth century [11,37]. This is also supported by the fact that few people over the age of 70 years in need of hearing aids use them [38], and the associated stigma of HL especially in the elderly may prevent them from using hearing aids [13].

Of concern based on the outcomes of this study is that more than one quarter of the adults in Jordan (27.6%) were not aware that prolonged exposure to noise using earphones may cause HL. This trend has also been observed among the population in Italy and Saudi [11,12]. On the other hand, adults surveyed in South Africa showed a high level of knowledge on the negative effect of noise on HL, which has been attributed to prevalent use of loud music in the villages and the awareness of the community about its impact. Binary logistic regression analysis did not reveal any association between the sample demographics and the inclination to provide a correct answer about the prolonged use of earphones. As the survey population was sampled from the main cities of Jordan, there may be an underrepresentation of the rural population which would justify the low level of knowledge related to noise. There is a need for health awareness campaigns in Jordan that include awareness about issues related to ear health and care, especially in the rural areas, and address health-related stigma, including stigma or misconceptions about hearing loss and ear health which would affect the public awareness and healthcare-seeking behavior [39]. In other disciplines of health such as dietary issues [40], and chronic non-communicable diseases [41], public health education has proven to enhance the level of knowledge about health issues and improve the attitudes of the population towards better health practices. This solidifies the need for concrete approach to public health awareness campaigns that either target ear health and care or include it in a more general health promotion agenda [42].

## Conclusions, limitations and recommendations

This study established the knowledge of the adult population in Jordan regarding hearing loss and hearing care. In terms of the external validity of the study, the results may not be strictly applicable to the elderly population as they only constituted around 8% of the sample population. Furthermore, as no teenagers were not included, this age group which may be exposed to loud noise may have a different level of knowledge about hearing loss and hearing care. Furthermore, as the study did not cover the rural areas of Jordan, it would be of interest to survey the population residing in these regions as their level of education and lifestyle may differ from persons residing in urban areas.

The current study would be of interest to educators, policymakers, ear, nose and throat specialists, and audiologists. It seems that there is a need for educators to address the knowledge

gap about ear heath and hearing in the curricula as part of the public health topics. Policy-makers are encouraged to introduce planned public awareness campaigns targeted to the at-risk populations such as the elderly and individuals with limited educational achievement, and men as they were the two population groups that received the lowest number of correct responses to the survey included in the study. Furthermore, public health campaigns could be tailored to address the misconceptions flagged in the current study such as: deaf–mute cannot speak because of defects in the vocal tract, cotton buds are necessary for ear cleaning and are the safest means, and that ear drops are sufficient to treat earache. Such awareness campaigns can be tailored for various population demographical groups to raise the level of awareness of the population which may alter their behaviors regarding ear care and attitudes towards hearing loss. Ear, nose and throat specialists, and audiologists need to play an active role in providing awareness about hearing and ear issues especially to men as women have demonstrated a higher level of knowledge about such issues.

One of the limitations of this study is the possible participation bias as only motivated subjects would have participated in the study. Also, as this study was questionnaire-based, thus recollection bias may have been introduced. One of the strengths of this study is the systematic sampling approach adopted to minimize sampling bias. In addition, inter-investigator bias was avoided by allowing one researcher to conduct the survey. Furthermore, the use of a validated questionnaire written in the language spoken by the participants minimized translation bias.

## Supporting information

**S1 Data.**
(XLSX)

## Author Contributions

**Conceptualization:** Yazan Gammoh, Rama Alasir.

**Data curation:** Yazan Gammoh, Rama Alasir, Laila Qanawati.

**Formal analysis:** Yazan Gammoh.

**Investigation:** Yazan Gammoh.

**Methodology:** Yazan Gammoh, Rama Alasir, Laila Qanawati.

**Project administration:** Yazan Gammoh.

**Resources:** Laila Qanawati.

**Software:** Yazan Gammoh.

**Supervision:** Yazan Gammoh.

**Validation:** Yazan Gammoh.

**Writing – original draft:** Yazan Gammoh.

**Writing – review & editing:** Yazan Gammoh, Rama Alasir, Laila Qanawati.

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
