## [Decision Letter · Decision Letter 0]

30 Sep 2024

PONE-D-24-05749Awareness of hearing loss and ear health among Jordanian adultsPLOS ONE

Dear Dr.  Gammoh,

Thank you for submitting your manuscript to PLOS ONE. After careful consideration, we feel that it has merit but does not fully meet PLOS ONE’s publication criteria as it currently stands. Therefore, we invite you to submit a revised version of the manuscript that addresses the points raised during the review process.

We look forward to receiving your revised manuscript.

Kind regards,

Ateya Megahed Ibrahim El-eglany

Academic Editor

PLOS ONE

Journal Requirements:

Reviewers' comments:

Reviewer's Responses to Questions

**Comments to the Author**

1. Is the manuscript technically sound, and do the data support the conclusions?

Reviewer #1: Yes

Reviewer #2: Yes

2. Has the statistical analysis been performed appropriately and rigorously? 

Reviewer #1: Yes

Reviewer #2: Yes

3. Have the authors made all data underlying the findings in their manuscript fully available?

Reviewer #1: Yes

Reviewer #2: Yes

4. Is the manuscript presented in an intelligible fashion and written in standard English?

Reviewer #1: Yes

Reviewer #2: Yes

5. Review Comments to the Author

Reviewer #1: The study investigates awareness of hearing loss among Jordanian adults. Specifically, it examines factors such as knowledge about hearing impairment, understanding of risk factors, and familiarity with available resources for managing hearing-related issues. Additionally, the study explores variations in awareness across different age groups, educational backgrounds, and hearing statuses. The findings contribute valuable insights to public health initiatives aimed at improving hearing health awareness in Jordan. However, the following comments are recommended for consideration,

Abstract:

1.Combine the method subsections in the abstract (design, participants, and main outcome and measures) into a single subsection titled “Methods.”

Introduction:

2. contextualize the study by referencing existing literature on awareness of hearing loss in Jordan and globally.

3. Highlight gaps in the current evidence base.

Methods: 4. Specify inclusion and exclusion criteria. 5. It is mentioned that the study questionnaire was adopted from the literature. One of the referenced studies has English origin. Address how the study questionnaire, adapted from literature, was translated to Arabic and validated. 6. Create a dedicated subsection describing the questionnaire’s content, the number of questions, their nature, and the required participant responses. 7. Provide the exclusion rate. 8. Define how participants were categorized as having normal or impaired hearing.

Results: 9. Explain why most participants had normal hearing. Acknowledge that ensuring representation of individuals with hearing loss or family history of hearing impairments can be challenging due to their lower numbers compared to those with normal hearing and no family history of hearing loss. 10. Report both numbers and percentages in the results. 11. Detail the training of personnel administering the instruments. 12. Address missing data and how it was handled in the analyses.

Discussion: 13. Add a title to the discussion section. 14. Discuss the robustness of the results. 15. Elaborate on educational levels and hearing loss incidences in Jordan and explain how the authors generalize their findings to the broader Jordanian population. 16. Explore the external validity of the study’s conclusions.

17. Compare the study’s results with related research on hearing loss awareness in Jordan, such as Alqudah et al.'s work (Parental knowledge and attitudes towards hereditary hearing loss and genetic technology).‏

Reviewer #2: 1. Title and Abstract

Areas for Improvement:

Title: The title is clear and concise but could be more engaging. Consider rephrasing to "Exploring Awareness of Hearing Loss and Ear Health in Jordanian Adults", to enhance its appeal to a broader audience.

Abstract: The abstract effectively summarizes the study, but it could benefit from adding more specific numerical results to highlight key findings (e.g., the percentage of respondents aware of certain facts). Additionally, mention the most common misconceptions identified in the study.

2. Introduction

Areas for Improvement:

Contextualization of Hearing Loss: The introduction should include more global data on the consequences of hearing loss, particularly its impact on quality of life, to provide broader context for the study.

Emphasize Preventive Strategies: While the introduction discusses the consequences of hearing loss, it could benefit from an expanded section on preventive strategies and how awareness can lead to early intervention.

3. Methods

Areas for Improvement:

Sampling Details: The systematic sampling method could be explained more thoroughly. For example, describe why every third person was chosen and whether this approach ensured a representative sample of the Jordanian population.

Statistical Justification: Expand the section on statistical analysis to provide more detail about how the choice of tests (e.g., t-tests, ANOVA) was justified given the study design.

4. Results

Areas for Improvement:

Interpretation of Findings: Although the results are well-presented, the discussion around gender differences in awareness could be expanded. Specifically, explain why women demonstrated higher levels of awareness in particular domains, linking it to cultural or societal roles in Jordan.

Detailed Misconceptions: Highlight the most concerning misconceptions more prominently in the results. For example, the incorrect belief about vocal tract defects causing deafness/muteness should be discussed in greater detail.

5. Discussion

Areas for Improvement:

Comparison with Other Studies: The discussion should draw more explicit comparisons between the results of this study and findings from similar studies in other Middle Eastern countries. This would help contextualize the findings and demonstrate the broader applicability of the results.

Link to Broader Health Education: Discuss how these findings could influence broader public health campaigns in Jordan. How might these results inform policy changes or educational interventions in schools or workplaces?

6. Conclusion

Areas for Improvement:

Actionable Recommendations: The conclusion could benefit from more specific recommendations for future public health campaigns, such as targeting certain misconceptions or focusing on at-risk populations (e.g., men, less educated individuals).

Future Research: Suggest areas for further research, particularly in rural populations or among different age groups, to identify whether similar trends in awareness are present across all demographics in Jordan.

7. References

Areas for Improvement:

Recent Studies: Include more recent references that focus on public awareness of hearing loss and ear health, particularly in developing countries. This will make the study’s context more relevant and up-to-date.

you may consider use the following recommendation to support your discussion :

Nashwan, Abdulqadir J., Valdez, Glenn Ford D., Villar, Ralph C., et al.

Stigma towards health care providers taking care of COVID-19 patients: A multi-country study.

Published: April 2022 in Heliyon

DOI: 10.1016/J.HELIYON.2022.E09300

This article explores health-related stigma, which can be used to highlight how stigma or misconceptions about hearing loss and ear health may affect public awareness and healthcare-seeking behavior.

Shaban, Marwa Mamdouh, Sharaa, Heba Magdy, Shaban, Mostafa, et al.

Effect of digital-based nursing intervention on knowledge of self-care behaviors and self-efficacy of adult clients with diabetes.

Published: February 2024 in BMC Nursing

DOI: 10.1186/S12912-024-01787-2

This study provides insights into how educational interventions improve health awareness, supporting the idea of public education campaigns for hearing loss and ear health.

Shaban, Mostafa, Mohammed, Huda Hamdy, Ibrahim, Ateya Megahed, et al.

Exploring the nurse-patient relationship in caring for the health priorities of older adults: qualitative study.

Published: July 2024 in BMC Nursing

DOI: 10.1186/S12912-024-02099-1

This article discusses the role of healthcare providers in raising patient awareness, which can relate to the role of audiologists and healthcare professionals in enhancing ear health awareness.

Bakhsh, Ebtisam, Shaban, Mostafa, AlSheef, Mohammed, et al.

Exploring the clinical efficacy of venous thromboembolism management in Saudi Arabian hospitals: An insight into patient outcomes.

Published: April 2023 in Journal of Personalized Medicine

DOI: 10.3390/JPM13040612

This article emphasizes the importance of effective management and patient awareness, which parallels the necessity for increased awareness and proper management of hearing loss in Jordan.

6. PLOS authors have the option to publish the peer review history of their article (what does this mean?). If published, this will include your full peer review and any attached files.

Reviewer #1: No

Reviewer #2: No

---

## [Author Response · Author response to Decision Letter 0]

12 Nov 2024

Editor:

Comment: 

Response:

Thank you. 

The affiliation of the authors was updated to match the journal’s formatting requirements. 

The colon (:) was removed from Corresponding author 

Reviewer 1:

1.Comment:

Combine the method subsections in the abstract (design, participants, and main outcome and measures) into a single subsection titled “Methods.”

Response:

Thank you for your comment, the change has been implemented

2.Comment:

contextualize the study by referencing existing literature on awareness of hearing loss in Jordan and globally.

Response: 

Thank you. Further references were cited in the introduction as suggested.

3.Comment:

Highlight gaps in the current evidence base.

Response: 

The gaps in relation to general population knowledge about HL and ear health have been highlighted

4. Comment:

Specify inclusion and exclusion criteria. 

Response: 

Inclusion and exclusion criteria were added.

5. Comment:

It is mentioned that the study questionnaire was adopted from the literature. One of the referenced studies has English origin. Address how the study questionnaire, adapted from literature, was translated to Arabic and validated.

Response: 

Thank you. This has been addressed in the newly created sub-section about the questionnaire. 

6. Comment:

Create a dedicated subsection describing the questionnaire’s content, the number of questions, their nature, and the required participant responses.

Response:

Thank you. The sub-section has been included.

7. Comment:

Provide the exclusion rate

Response: 

Thank you. This has been included in the methods section.

8. Comment:

Define how participants were categorized as having normal or impaired hearing.

Response: 

This has been defined in table 3 in the results section and in the discussion section.

9. Comment:

Explain why most participants had normal hearing. Acknowledge that ensuring representation of individuals with hearing loss or family history of hearing impairments can be challenging due to their lower numbers compared to those with normal hearing and no family history of hearing loss.

Response:

Thank you. This has been addressed in the discussion section.

10. Comment:

Report both numbers and percentages in the results.

Response: 

Done throughout the results section

11. Comment:

Detail the training of personnel administering the instruments.

Response:

This information has been added in the methods section.

12. Comment:

Address missing data and how it was handled in the analyses.

Response:

This now has been addressed in the methods section.

13. Comment:

Add a title to the discussion section.

Response:

A title to the discussion section was added.

14. Comment:

Discuss the robustness of the results.

Response:

This now has been addressed in the discussion.

15. Comment:

Elaborate on educational levels and hearing loss incidences in Jordan and explain how the authors generalize their findings to the broader Jordanian population.

Response:

Thank you. More details about educational levels and generalizing the findings were added in the discussion.

16. Comment: 

Explore the external validity of the study’s conclusions.

Response:

The external validity has been addressed in the conclusion section.

17. Comment:

Compare the study’s results with related research on hearing loss awareness in Jordan, such as Alqudah et al.'s work (Parental knowledge and attitudes towards hereditary hearing loss and genetic technology).

Response:

Thank you. That reference has been added and related research has been discussed.

Reviewer 2:

1. Comment:

Title: The title is clear and concise but could be more engaging. Consider rephrasing to "Exploring Awareness of Hearing Loss and Ear Health in Jordanian Adults", to enhance its appeal to a broader audience.

Abstract: The abstract effectively summarizes the study, but it could benefit from adding more specific numerical results to highlight key findings (e.g., the percentage of respondents aware of certain facts). Additionally, mention the most common misconceptions identified in the study.

Response:

Thank you for your suggestion regarding the title, it has been changed accordingly

The abstract has been updated as per the recommendations. 

2. Comment:

Contextualization of Hearing Loss: The introduction should include more global data on the consequences of hearing loss, particularly its impact on quality of life, to provide broader context for the study.

Emphasize Preventive Strategies: While the introduction discusses the consequences of hearing loss, it could benefit from an expanded section on preventive strategies and how awareness can lead to early intervention.

Response: 

Thank you. The recommended additions have been implemented in the introduction section.

3. Comment:

Sampling Details: The systematic sampling method could be explained more thoroughly. For example, describe why every third person was chosen and whether this approach ensured a representative sample of the Jordanian population.

Statistical Justification: Expand the section on statistical analysis to provide more detail about how the choice of tests (e.g., t-tests, ANOVA) was justified given the study design.

Response: 

Thank you. The explanation about the sampling method has been added, as well as the justification for using the statistical tests.

4. Comment:

Interpretation of Findings: Although the results are well-presented, the discussion around gender differences in awareness could be expanded. Specifically, explain why women demonstrated higher levels of awareness in particular domains, linking it to cultural or societal roles in Jordan.

Detailed Misconceptions: Highlight the most concerning misconceptions more prominently in the results. For example, the incorrect belief about vocal tract defects causing deafness/muteness should be discussed in greater detail.

Response:

Thank you for highlighting this point. The discussion has been updated, specifically about women.

More details about misconceptions have been discussed.

5. Comment:

Comparison with Other Studies: The discussion should draw more explicit comparisons between the results of this study and findings from similar studies in other Middle Eastern countries. This would help contextualize the findings and demonstrate the broader applicability of the results.

Link to Broader Health Education: Discuss how these findings could influence broader public health campaigns in Jordan. How might these results inform policy changes or educational interventions in schools or workplaces?

Response:

The recommended comparisons, the broader applicability and link to broader health education have now been addressed.

6. Comment:

Actionable Recommendations: The conclusion could benefit from more specific recommendations for future public health campaigns, such as targeting certain misconceptions or focusing on at-risk populations (e.g., men, less educated individuals).

Future Research: Suggest areas for further research, particularly in rural populations or among different age groups, to identify whether similar trends in awareness are present across all demographics in Jordan.

Response:

Thank you. The recommendations were included in the conclusion section and within the discussion section.

7. Comment:

Recent Studies: Include more recent references that focus on public awareness of hearing loss and ear health, particularly in developing countries. This will make the study’s context more relevant and up-to-date.

Response:

Thank you for your valuable suggestions. All the suggested references were cited in the discussion section and have been added as references: 29, 35, 39,42

---

## [Decision Letter · Decision Letter 1]

16 Dec 2024

Exploring awareness of hearing loss and ear health in Jordanian adults

PONE-D-24-05749R1

Dear Authors,

We’re pleased to inform you that your manuscript has been judged scientifically suitable for publication and will be formally accepted for publication once it meets all outstanding technical requirements.

Kind regards,

Ateya Megahed Ibrahim El-eglany

Academic Editor

PLOS ONE

Additional Editor Comments (optional):

Reviewers' comments:

Reviewer's Responses to Questions

**Comments to the Author**

1. If the authors have adequately addressed your comments raised in a previous round of review and you feel that this manuscript is now acceptable for publication, you may indicate that here to bypass the “Comments to the Author” section, enter your conflict of interest statement in the “Confidential to Editor” section, and submit your "Accept" recommendation.

Reviewer #1: All comments have been addressed

Reviewer #2: All comments have been addressed

2. Is the manuscript technically sound, and do the data support the conclusions?

Reviewer #1: Yes

Reviewer #2: Yes

3. Has the statistical analysis been performed appropriately and rigorously? 

Reviewer #1: Yes

Reviewer #2: Yes

4. Have the authors made all data underlying the findings in their manuscript fully available?

Reviewer #1: Yes

Reviewer #2: Yes

5. Is the manuscript presented in an intelligible fashion and written in standard English?

Reviewer #1: Yes

Reviewer #2: (No Response)

6. Review Comments to the Author

Reviewer #1: Dear Editor,

I have reviewed the revised manuscript titled "Exploring Awareness of Hearing Loss and Ear Health in Jordanian Adults" by Yazan Gammoh. I am pleased to report that the authors have addressed all my previous comments and concerns and have done so with thoroughness and commitment, significantly improving the manuscript's quality.

Overall Quality and Contribution:

The manuscript now presents a significant contribution to the field of Audiology in Jordan. The research question is well-defined, and the methodology is robust.

Clarity and Organization:

The manuscript is well-organized and easy to follow. The introduction and discussion sections have been improved for better clarity and impact.

Literature Review:

The literature review now includes recent studies on the awareness of hearing loss in Jordan, providing a current perspective.

Methodology:

The methodology is robust, with a detailed justification for the choice of method and a clear description of variables and data collection.

Results and Discussion:

The results are well-presented, with clearly labeled figures and tables. The discussion includes a deeper analysis of the factors that impacted the results and addresses potential limitations.

Ethical Considerations:

All ethical guidelines have been followed, with no apparent concerns.

Dual Publication:

The manuscript appears to be original and unpublished elsewhere.

Overall, this is well-written and prepared for publication.

Reviewer #2: thank you for the author for completeing all revision requested , the paper is ready for publication in the current form

7. PLOS authors have the option to publish the peer review history of their article (what does this mean?). If published, this will include your full peer review and any attached files.

Reviewer #1: No

Reviewer #2: **Yes: **Mostafa Shaban

---

## [Editor Report · Acceptance letter]

17 Dec 2024

PONE-D-24-05749R1 

PLOS ONE

Dear Dr. Gammoh, 

I'm pleased to inform you that your manuscript has been deemed suitable for publication in PLOS ONE. Congratulations! Your manuscript is now being handed over to our production team.

Kind regards, 

on behalf of

Dr. Ateya Megahed Ibrahim El-eglany 

Academic Editor

PLOS ONE